# Identification of a Toxin–Antitoxin System That Contributes to Persister Formation by Reducing NAD in *Pseudomonas aeruginosa*

**DOI:** 10.3390/microorganisms9040753

**Published:** 2021-04-02

**Authors:** Jingyi Zhou, Shouyi Li, Haozhou Li, Yongxin Jin, Fang Bai, Zhihui Cheng, Weihui Wu

**Affiliations:** State Key Laboratory of Medicinal Chemical Biology, Key Laboratory of Molecular Microbiology and Technology of the Ministry of Education, Department of Microbiology, College of Life Sciences, Nankai University, Tianjin 300071, China; 2120181072@mail.nankai.edu.cn (J.Z.); lishouyi@mail.nankai.edu.cn (S.L.); 2120191011@mail.nankai.edu.cn (H.L.); yxjin@nankai.edu.cn (Y.J.); baifang1122@nankai.edu.cn (F.B.); zhihuicheng@nankai.edu.cn (Z.C.)

**Keywords:** *Pseudomonas aeruginosa*, persister, toxin/antitoxin system

## Abstract

Bacterial persisters are slow-growing or dormant cells that are highly tolerant to bactericidal antibiotics and contribute to recalcitrant and chronic infections. Toxin/antitoxin (TA) systems play important roles in controlling persister formation. Here, we examined the roles of seven predicted type II TA systems in the persister formation of a *Pseudomonas aeruginosa* wild-type strain PA14. Overexpression of a toxin gene *PA14_51010* or deletion of the cognate antitoxin gene *PA14_51020* increased the bacterial tolerance to antibiotics. Co-overexpression of *PA14_51010* and *PA14_51020* or simultaneous deletion of the two genes resulted in a wild-type level survival rate following antibiotic treatment. The two genes were located in the same operon that was repressed by PA14_51020. We further demonstrated the interaction between PA14_51010 and PA14_51020. Sequence analysis revealed that PA14_51010 contained a conserved RES domain. Overexpression of *PA14_51010* reduced the intracellular level of nicotinamide adenine dinucleotide (NAD^+^). Mutation of the RES domain abolished the abilities of PA14_51010 in reducing NAD^+^ level and promoting persister formation. In addition, overproduction of NAD^+^ by mutation in an *nrt*R gene counteracted the effect of *PA14_51010* overexpression in promoting persister formation. In combination, our results reveal a novel TA system that contributes to persister formation through reducing the intracellular NAD^+^ level in *P. aeruginosa*.

## 1. Introduction

Persister cells are dormant cells that are highly tolerant to environmental stresses, such as nutrient starvation, oxidative stress and antibiotics [1,2,3,4]. The formation of persister cells is mainly due to phenotype switch [5]. When the environmental stress is removed the dormant persister cells will resume growth [5]. Persister cells might be a major cause of chronic and recurrent bacterial infections. In addition, persister cells can be a reservoir for the evolution of antibiotic-resistant mutants [6,7,8].

In bacteria, toxin/antitoxin (TA) systems play important roles in persister formation [9,10,11,12]. A TA system is composed of a stable toxin and a labile antitoxin. The toxin represses bacterial growth by inhibiting important bacterial physiological processes, such as DNA replication, transcription, protein synthesis, cell wall synthesis, cell division or reducing membrane potential [13]. Normally, the function of a toxin is neutralized by its cognate antitoxin. It is proposed that the antitoxin is degraded stochastically or in response to environmental stresses, which leads to the activation of the toxin. Based on the natures of the antitoxins and their mechanisms of action, the TA systems have been classified into six types (I–VI) [9]. Recently, a group of type VII TA systems was discovered in *Escherichia coli* [14], *Yersinia enterocolitica* [15], *Mycobacterium tuberculosis* [16] and *Shewanella oneidensis* [17,18]. The type II TA systems are widespread in bacterial genomes [19]. For example, at least 79 and 12 type II TA systems were identified in *M. tuberculosis* and *E. coli*, respectively [20,21]. A typical type II TA system is composed of two genes in an operon that encode a toxin protein and an antitoxin protein. The activity of a type II toxin is inhibited by direct binding to its cognate antitoxin and the transcription of the TA gene operon is repressed by the antitoxin. 

*Pseudomonas aeruginosa* is a Gram-negative opportunistic pathogen that causes acute and chronic infections in immunocompromised patients, such as those suffering AIDS or undergoing chemotherapy [22]. In addition, *P. aeruginosa* commonly causes pulmonary infections in cystic fibrosis (CF) patients, promoting an accelerated decline of lung function. The mucociliary deficiency in the CF airway induces the establishment of an environment which promotes *P. aeruginosa* growth [22]. It was demonstrated that the persister formation was increased in the biofilm of *P. aeruginosa* [23,24]. The roles of the TA systems in persister formation were explored in *P. aeruginosa*. At least four pairs of type II TA systems were identified in *P. aeruginosa*, namely ParD/ParE [25], HicA/HicB [26], RelE/RelB [27] and HigB/HigA [28]. The toxin ParE directly binds to and inhibits the function of the DNA gyrase [25]. The toxin HigB functions as an Rnase [28]. The deletion of the antitoxin gene *higA* reduces swarming motility and the production of pyochelin and pyocyanin. We previously demonstrated that the expression of the *higA*–*higB* operon was induced by ciprofloxacin. The overexpression of the toxin gene *higB* increases persister formation under ciprofloxacin treatment [29]. In addition, higB reduces the intracellular c-di-GMP level by activating the expression of three c-di-GMP hydrolysis genes, which increases expression of the type III secretion system genes and decreases biofilm formation [30]. Besides repressing the promoter of the *higA*–*higB* operon, higA was found to directly bind to and repress the promoter of *mvfR*, a key regulatory gene of the quorum-sensing system [31]. These results demonstrated the roles of both the toxin and the antitoxin in manipulating bacterial gene expression, besides affecting the bacterial growth and persister formation. A previous bioinformatics analysis identified seven additional putative type II TA systems on the genome of a wild-type *P. aeruginosa* stain PA14, namely PA14_51010/PA14_51020, PA14_40220/PA14_40210, PA14_21710/PA14_21720, PA14_28790/PA14_28780, PA14_60050/PA14_60040, PA14_71340/PA14_71330 and PA14_28120/PA14_28130 [20]. 

In this study, we examined the roles of these predicted type II TA systems in persister formation. PA14_51010/PA14_51020 (PA1030/PA1029 on the PAO1 genome) was found to modulate persister formation by reducing the intracellular NAD^+^ level. Our results reveal a novel TA system involved in the persister formation by influencing the key molecule in energy metabolism in *P. aeruginosa*.

## 2. Materials and Methods

### 2.1. Strains and Plasmids

The strains, plasmids and primers used in this study are listed in Table 1. All strains were cultured in the Luria–Bertani (LB) broth (10 g/L tryptone, 5 g/L yeast extract, 5 g/L NaCl, pH 7.4) (BBI LifeScience, Shanghai, China) at 37 °C. When needed, antibiotics were used at the following concentrations (μg/mL): for *E. coli*, ampicillin 100, gentamicin 10 and tetracycline 10; for *P. aeruginosa*, carbenicillin 150, gentamicin 50 and tetracycline 50 (Macklin, Shanghai, China). Isopropyl-β-d-thiogalactoside (IPTG) (TaKaRa, Dalian, China) was added to LB at indicated concentrations.

### 2.2. Construction of Plasmids and Mutation Strains

For the overexpression of *PA14_51010*, the *PA14_51010* coding region with its natural Shine–Dalgarno (SD) sequence was amplified by PCR using the PA14 genomic DNA as the template with the primers listed in Table 1. The PCR product was cloned into the plasmid pMMB67EH. For the overexpression of *PA14_51020*, the *PA14_51020* coding region was amplified by PCR using the PA14 genomic DNA as the template and cloned into pMMB67EH containing the *gst* gene [36], resulting in an N-terminus GST tagged *PA14_51020*. 

To construct the transcriptional fusion of the *PA14_51020* promoter and the *lac*Z gene, a 500 bp DNA fragment upstream of the *PA14_51020* gene start codon was amplified by PCR with the primers listed in Table 1. The PCR product was cloned into the pUCP20-*lac*Z plasmid [35].

For the deletion of the *PA14_51020* gene, a 986-bp fragment and a 1019-bp fragment upstream and downstream of the *PA14_51020* gene coding region were amplified by PCR using the PA14 genomic DNA as the template. The PCR products were ligated into the pEX18Tc plasmid. Deletion of the *PA14_51020* gene in *P. aeruginosa* was performed as previously described [37]. The same method was used to construct the Δ*PA14_51010* and Δ*PA14_51010*Δ*PA14_51020* mutants.

### 2.3. Bacteria Killing Assay

Overnight bacterial cultures of the indicated strains were diluted 100-fold in fresh LB. To induce the expression of the cloned genes, 1 mM IPTG (TaKaRa, Dalian, China) was added to the culture medium. The bacteria were cultured to an OD_600_ of 0.8–1.0, and then treated with 0.5 μg/mL ciprofloxacin (BBI LifeScience, Shanghai, China) or 5 μg/mL tobramycin (Meilunbio, Dalian, China). At indicated time points, the bacterial samples were taken for serial dilution and plating. The plate was incubated at 37 °C for 24 h before colony counting.

### 2.4. RNA Isolation and Quantitative Real-Time PCR

Total bacterial RNA was isolated with the ZOMANBIO RNA Rapid Extraction Kit (Zomanbio, Beijing, China). A Prime-Script Reverse Transcriptase (TaKaRa, Dalian, China) and random primers were used to synthesize cDNA from each RNA sample as instructed by the manufacturer. Quantitative RT-PCR was performed with a CFX real-time system (Bio-Rad, Hercules, CA, USA), using SYBR Premix ExTaq II (TaKaRa, Dalian, China) and indicated primers (Table 1). The 30 S subunit ribosomal protein coding gene *rpsL* was used as an internal control.

### 2.5. Transcriptional Reporter Assay

Bacteria of the indicated strains were grown to an OD_600_ of 1.0 in LB at 37 °C. 0.5 mL of the bacterial culture was subjected to centrifugation at 12,000× *g* for 1 min. The bacteria were resuspended in 1.5 mL Z buffer (60 mM Na_2_HPO_4_, 60 mM NaH_2_PO_4_, 1 mM MgSO_4_, 10 mM KCl, 50 mM β-mercaptoethanol (pH 7.0); BBI Life Sciences, Shanghai, China). The β-galactosidase activity was determined by ortho-nitrophenyl- galactopyranoside (ONPG) (BBI Life Sciences, Shanghai, China) as described previously [38].

### 2.6. Expression and Purification of the PA14_51020 Protein

DH5α cells carrying the pMMB67EH-*gst-PA14_51020*-His were grown in LB at 37 °C. When the OD_600_ reached approximately 0.6, 1 mM IPTG was added to the medium and the bacteria were kept growing for 4 h at 37 °C. The bacteria were collected by centrifugation at 6000× *g* for 10 min at 4 °C, followed by resuspension in the lysis buffer (20 mM Tris-HCl, 150 mM NaCl, 3 mM β-mercaptoethanol, 10 mM imidazole, 0.5% NP-40 (Solarbio, Beijing, China), pH 8.0) and sonication. The lysate was centrifuged at 12,000 *g* for 10 min at 4 °C. The supernatant was incubated with Ni-NTA beads (Qiagen, Düsseldorf, North Rhine-Westphalia, GER) for 2 h at room temperature. The Ni-NTA beads were washed sequentially with the lysis buffer containing 50 mM and 100 mM imidazole. The bound proteins were eluted by the lysis buffer containing 300 mM imidazole. The purity of the eluted protein was examined by SDS-PAGE, and the protein concentrations were quantified by a bicinchoninic acid (BCA) protein assay (Beyotime, Shanghai, China).

### 2.7. Electrophoretic Mobility Shift Assay (EMSA)

The EMSA was performed as previously described [39]. A 200 bp DNA fragment upstream of the *PA14_51020* coding region or inside of the *PA14_51010* coding region was amplified by PCR with primers listed in Table 1. An amount of 20 ng of the purified PCR product was incubated with the purified GST-PA14_51020-His protein at the concentration of 8, 16 or 32 mM in the binding buffer (10 mM Tris–HCl, 5 mM CaCl_2_, 100 mM NaCl, 4% glycerol (promega, Beijing, China), 1 mM EDTA (solarbio, Beijing, China), 10 mM DTT (solarbio, Beijing, China), pH 7.5) in a total volume of 20 μL. The mixture was incubated on ice for 15 min. The purified GST protein was used as a negative control. The samples were loaded onto an 8% native polyacrylamide gel in 0.5 × Tris-borate-EDTA (TBE) buffer that had been prerun on ice for one hour. The electrophoresis was performed on ice at 120 V for 110 min, followed by staining in 0.5 × TBE containing 2 μg/mL ethidium bromide. The bands were observed with a molecular imager ChemiDoc™ XRS+ (Bio-Rad, Hercules, CA, USA).

### 2.8. Pull Down Assay

The GST-His and GST-PA14_51020-His proteins were purified by the Ni-NTA beads (Qiagen, Düsseldorf, North Rhine-Westphalia, GER) as aforementioned. The *E. coli* strain containing the pMMB67EH-*PA14_51010*-Flag was grown to an OD_600_ of 0.6 and then kept growing with 1 mM IPTG for 4 h at 37 °C. 600 mL of the bacteria was collected by centrifugation at 10,000 *g* for 1 min and resuspended in 12 mL lysis buffer (20 mM Tris-HCl, 150 mM NaCl, 3 mM β-mercaptoethanol, 10 mM imidazole, 0.5% NP-40, pH 8.0). An amount of 100 μg of the purified GST-His or GST-PA14_51020-His proteins was mixed with 4 mL of the bacterial lysate, followed by incubation with the Ni-NTA beads at 4 °C overnight. The Ni-NTA beads were then washed 5 times with the lysis buffer containing 50 mM imidazole. The proteins were eluted with the lysis buffer containing 300 mM imidazole. The samples were examined with an Anti-GST antibody (Applygen, Beijing, China) and an Anti-Flag antibody (Sigma, St. Louis, MO, USA).

## 3. Results

### 3.1. Identification of Novel Persister Formation Related Genes

In order to examine the roles of the predicted TA systems in persister formation, we overexpressed the potential toxin genes in wild-type PA14, including *PA14_28120*, *PA14_21710*, *PA14_28790*, *PA14_40220*, *PA14_51010*, *PA14_60050* and *PA14_71340*. Previously, we determined the minimum inhibitory concentrations (MICs) of tobramycin and ciprofloxacin against the wild-type PA14 as 1 and 0.25 μg/mL, respectively [29,36]. Overexpression of *PA14_51010* increased the bacterial survival rate by approximately 100-fold upon treatment with 5 μg/mL tobramycin or 0.5 μg/mL ciprofloxacin (Figure 1A,B). Overexpression of *PA14_28120* increased the survival rate by approximately 10-fold after the bacteria were treated with tobramycin for 8 h (Figure 1A), whereas overexpression of the other five genes did not affect the bacterial survival rates under ciprofloxacin treatment (Figure 1A,B). Notably, overexpression of *PA14_21710*, *PA14_28790*, *PA14_40220* and *PA14_60050* reduced the survival rates 2, 4 or 6 h after tobramycin treatment. However, the survival rates of strains overexpressing these genes were similar to PA14 containing the empty vector after 8 h (Figure 1A). These results suggest an important role of *PA14_51010* in persister formation, thus we focused our following studies on this gene.

### 3.2. PA14_51020 Regulates the Operon of PA14_51020 and PA14_51010

The type II toxin and antitoxin genes usually form one operon that is negatively regulated by the antitoxin. On the PA14 genome, *PA14_51010* is located next to *PA14_51020* and the two genes are transcribed in the same direction (Figure 2A) [22]. RT-PCR analysis confirmed that *PA14_51020* and *PA14_51010* were in the same operon (Figure 2B). To examine whether PA14_51020 negatively regulates the expression of the operon, we deleted the *PA14_51020* gene in wild-type PA14. The mRNA level of *PA14_51010* was higher in the Δ*PA14_51020* mutant than that in wild-type PA14 (Figure 2C). We then constructed a transcriptional fusion of *lac*Z with the promoter of the operon (designated as P*_PA14_51020-_lac*Z) and transferred it into wild-type PA14 and the Δ*PA14_51020* mutant. The β–galactosidase activity was higher in the Δ*PA14_51020* mutant than that in wild-type PA14 (Figure 2D). In addition, an electrophoretic mobility shift assay (EMSA) demonstrated an interaction between PA14_51020 and the promoter region of the operon but not the fragment inside the coding region of *PA14_51010* (Figure 2E). These results suggest that *PA14_51020* and *PA14_51010* are in the same operon, which is directly repressed by PA14_51020.

### 3.3. PA14_51020 and PA14_51010 Regulates Persister Formation

To verify that PA14_51020 and PA14_51010 function as a toxin–antitoxin system, we examined the interaction between PA14_51020 and PA14_51010 by a pull-down assay. The lysate of DH5α cells overexpressing *PA14_51010*-Flag was incubated with purified 6×His tagged GST (GST-His) or GST-PA14_51020 (GST-PA14_51020-His). In the eluted protein samples, there was a higher amount of GST-His than GST-PA14_51020-His, indicating that GST-PA14_51020-His might be more labile than GST-His in the experimental process. Nevertheless, the PA14_51010-Flag was co-purified with GST-PA14_51020-His but not GST-His, indicating an interaction between PA14_51010 and PA14_51020 (Figure 3). These result indicate that PA14_51010 and PA14_51020 could function as a type II TA system. 

We then examined whether PA14_51020 could suppress the function of PA14_51010 in promoting persister formation. Compared to the strain overexpressing *PA14_51010*, co-expressing *PA14_51010* and *PA14_51020* reduced the bacterial survival rates to the wild-type level following treatment with ciprofloxacin and tobramycin (Figure 4A,B). In wild-type PA14, deletion of *PA14_51020* increased the bacterial survival rate by approximately 60-fold following the treatment of tobramycin, whereas simultaneous deletion of *PA14_51010* and *PA14_51020* reduced the survival rate to the wild-type level (Figure 4C). However, the survival rate of the Δ*PA14_51020* mutant was similar to the wild-type strain under the treatment of ciprofloxacin (Figure 4D). To understand the discrepancy between the *PA14_51010* overexpressing strain and the Δ*PA14_51020* mutant, we determined the expression levels of *PA14_51010* by real time PCR. The mRNA level of *PA14_51010* was approximately 5-fold lower in the Δ*PA14_51020* mutant than that in the *PA14_51010* overexpressing strain (Figure 4E), which might result in the different survival rates in the two strains. In combination, these results demonstrate that PA14_51020 and PA14_51010 comprise a toxin and antitoxin system that might play a more important role in promoting persister formation under the treatment of tobramycin than ciprofloxacin.

### 3.4. PA14_51010 Promotes Persister Formation by Reducing the Intracellular Level of NAD^+^

PA14_51010 contains a conserved RES domain (Figure 5A). Overexpression of *PA14_51010* did not reduce the bacterial growth rate in LB (Appendix A). On the PA14 genome, the start codon of *PA14_51010* is GTG [40], which is the same as the orthologues genes (designated as *res*) in *P. luminescens*, *M. tuberculosis* and *Pseudomonas putida* [41]. A previous study in *P. luminescens* demonstrated that overexpression of the *res* gene did not affect the bacterial growth rate either [41]. However, substitution of the GTG with ATG reduced the growth rate, which might be due to the enhanced translation of the *res* gene [41]. Consistent with the previous study, the replacement of the start codon of the *PA14_51010* gene with ATG increased the protein level and reduced the bacterial growth rate (Appendix A). 

Previous studies revealed NAD^+^ degradation activities of the bacterial RES toxins [41,42]. Overexpression of *PA14_51010* in wild-type PA14 indeed reduced the intracellular level of NAD^+^, which was restored by co-expression of *PA14_51020* (Figure 5B). Simultaneous replacement of the R, E and S residues with A residues did not reduce the level of protein PA14_51010-Flag (Figure 5C), but abolished the NAD^+^-reducing (Figure 5B) and persister formation-promoting activity of PA14_51010 (Figure 5D,E).

If PA14_51010 promotes persister formation by reducing NAD^+^, overproduction of NAD^+^ should neutralize the function of PA14_51010. In most bacteria, NAD^+^ is synthesized through a de novo synthesis pathway and salvage pathways I and/or II [43]. A conserved transcriptional regulator NrtR controls the expression of genes involved in NAD^+^ biosynthesis [44]. In *Mycobacterium smegmatis*, mutation of the *nrt*R gene increases the expression of NAD^+^ biosynthesis genes and intracellular NAD^+^ level [44]. In *P. aeruginosa*, NrtR directly represses genes involved in the NAD^+^ synthesis salvage pathway I, including *pncA*, *pncB1*, *nadE* and *nadD2* [45]. A previous study demonstrated that mutation of the *nrtR* gene results in upregulation of these genes and higher ratios of NAD^+^ to NADH and NADP^+^ to NADPH at the stationary growth phase [45]. We found that the total NAD^+^ level was increased in an *nrtR*::Tn mutant at the exponential growth phase (OD_600_ = 1.0) (Figure 6A). Overexpression of *PA14_51010* in the *nrtR*::Tn mutant reduced the NAD^+^ level, which however was higher than that in the wild-type PA14 (Figure 5C, Figure 6A). Upon treatment with tobramycin, the survival rate of the *nrtR*::Tn mutant containing the empty vector or overexpressing *PA14_51010* were higher than that of the wild-type strain containing the empty vector at the 6-h time point, but reduced to the wild-type level after 8 h (Figure 6B). In the presence of ciprofloxacin, overexpression of *PA14_51010* in the *nrtR*::Tn mutant increased the survival rate at the 4- and 6-h time points, but resulted in a similar survival rate as the strains containing the empty vector at the 8-h time point (Figure 6C). In contrast, overexpression of *PA14_51010* in the wild-type strain steadily increased the survival rates (Figure 6B,C). Taken together, these results demonstrate that PA14_51010 promotes persister formation by reducing the intracellular level of NAD^+^.

## 4. Discussion

The developing antibiotic resistance of *P. aeruginosa* imposes a severe threat on human health all over the world [46,47,48,49]. It has been demonstrated that bacterial persister cells are able to form a reservoir for the development of antibiotic-resistant mutants [7,8]. TA systems have been shown to be involved in persister formation [9,10,11,12]. In this study we screened the predicted type II TA systems in *P. aeruginosa* wild-type strain PA14 and identified that PA14_51010-PA14_51020 modulates persister formation. PA14_51010 belongs to the RES family protein and our results reveal that it promotes persister formation by reducing the intracellular NAD^+^ level. In *Photorhabdus luminescens*, *Mycobacterium tuberculosis* and *Sinorhizobium meliloti*, the cognate antitoxin of the RES toxin is named Xre (Xenobiotic response element) [43,50]. Based on the active domain and physiological function of PA14_51010 (PA1030 in PAO1), we propose to annotate it as ResA and the cognate antitoxin PA14_51020 (PA1029 in PAO1) as XreA. 

NAD^+^ is an electron carrier that participates in biological respiration and redox reactions and functions as a cofactor by NAD^+^-consuming enzymes. Therefore, reducing the intracellular level of NAD^+^ by ResA might rewire the cell metabolism and interfere with the redox homeostasis, which might contribute to increased antibiotic tolerance. A recent study demonstrated that overexpression of a *M. tuberculosis* toxin MbcT in *Mycobacterium smegmatis* depleted the intracellular NAD+ and lead to cell death [51]. We suspect that ResA might display obvious growth inhibitory effects under certain environmental stresses such as carbon or amino acid starvation. It warrants further study to examine the conditions that trigger the expression and activation of ResA. 

Thus far, at least five type II TA systems have been identified in *P. aeruginosa*, including ParD/ParE, HicA/HicB, RelE/RelB, HigB/HigA and ResA/XreA. ParE inhibits the function of DNA gyrase [25]. HigB has been found to cleave mRNA, which represses cell growths. Previous studies revealed that the toxin HigB affected swarming motility, pyochelin production, biofilm formation and expression of the type III secretion system (T3SS) genes, as well as numerous other genes in *P. aeruginosa* [28,29,30]. We demonstrated that HigB positively controlled three c-di-GMP hydrolysis genes. Thus, overexpression of *higB* reduces the intracellular level of c-di-GMP, resulting in upregulation of the T3SS genes and a reduction in biofilm formation [30]. Meanwhile, the cognate antitoxin HigA modulates the quorum-sensing systems by controlling the expression of MvfR independent of HigB [31]. It might be interesting to examine the roles of ResA/XreA in modulating global gene expression as well as the metabolic pathways. 

Previously, we examined the gene expression profile of *P. aeruginosa* persister cells and found an operon composed of PA2282-PA2287 was upregulated. In the operon, PA2285 and PA2287 promote persister formation. We further demonstrated that PA2285 and PA2287 suppress transcription and cell division by directly binding to RNA polymerase and FtsZ [52]. 

Here, we demonstrated that the overexpression of *resA* in wild-type PA14 increased the persister formation under tobramycin or ciprofloxacin treatment; however, deletion of the chromosomal *resA* did not affect the persister formation level. In *E. coli* and *S. typhimurium*, at least 37 TA systems have been identified [53]. Deletion of one of the TA systems did not affect the overall persister formation, presumably due to the redundancy of the TA systems [54,55]. Goormaghtigh et al. demonstrated that simultaneous deletion of 10 type II TA systems in *E. coli* did not affect the persister formation level under ampicillin or ofloxacin treatment [56]. It was demonstrated in *Salmonella enterica* that 14 type II TA systems were activated by nutrient starvation, phagocytosis by macrophages and transient acidification, which promoted persister formation [57]. However, simultaneous deletion of three TA systems (ecnB, phd-doc, and shpAB) did not affect persister formation [58]. These results suggest that additional TA systems or other determinants are involved in persister formation, which might compensate for the loss of one or multiple TA systems.

The alarmone molecule guanosine pentaphosphate/tetraphosphate, (p)ppGpp, has been shown to control the expression of TA systems and play important roles in persister formation as well as the bacterial response to environmental stresses, such as oxidative stress, starvation, etc. [59,60]. Meanwhile, it has been demonstrated that (p)ppGpp controls global gene expression and contributes to bacterial virulence in various infection models [61,62,63]. In *P. aeruginosa*, the role of (p)ppGpp in the regulation of the TA systems remains unknown. We previously demonstrated that the expression of *higA* was induced by ciprofloxacin treatment [29]. To further understand the roles of (p)ppGpp in the regulation of the identified TA systems and the bacterial response to antibiotics it is necessary to examine the transcriptomic profiles in wild-type and (p)ppGpp-deficient strains in the presence and absence of antibiotics. 

Overall, our study revealed a novel type II TA system that modulated persister formation in *P. aeruginosa*.

## Figures and Tables

**Figure 1 microorganisms-09-00753-f001:**
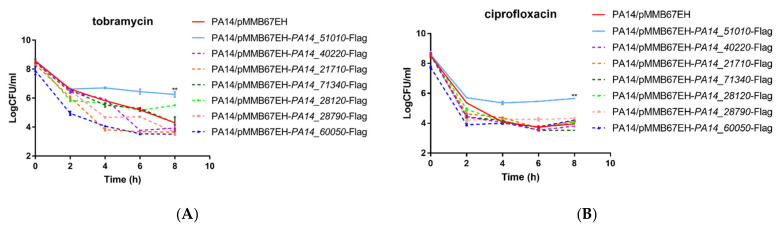
Identification of novel persister formation related genes. Wild-type PA14 overexpressing the indicated genes were treated with 5 µg/mL tobramycin (**A**) or 0.5 µg/mL ciprofloxacin (**B**). At indicated time points the live bacteria numbers were determined by serial dilution and plating. Data represent the results from three independent experiments. **, *p* < 0.01 compared to the other samples by Student’s *t* test.

**Figure 2 microorganisms-09-00753-f002:**
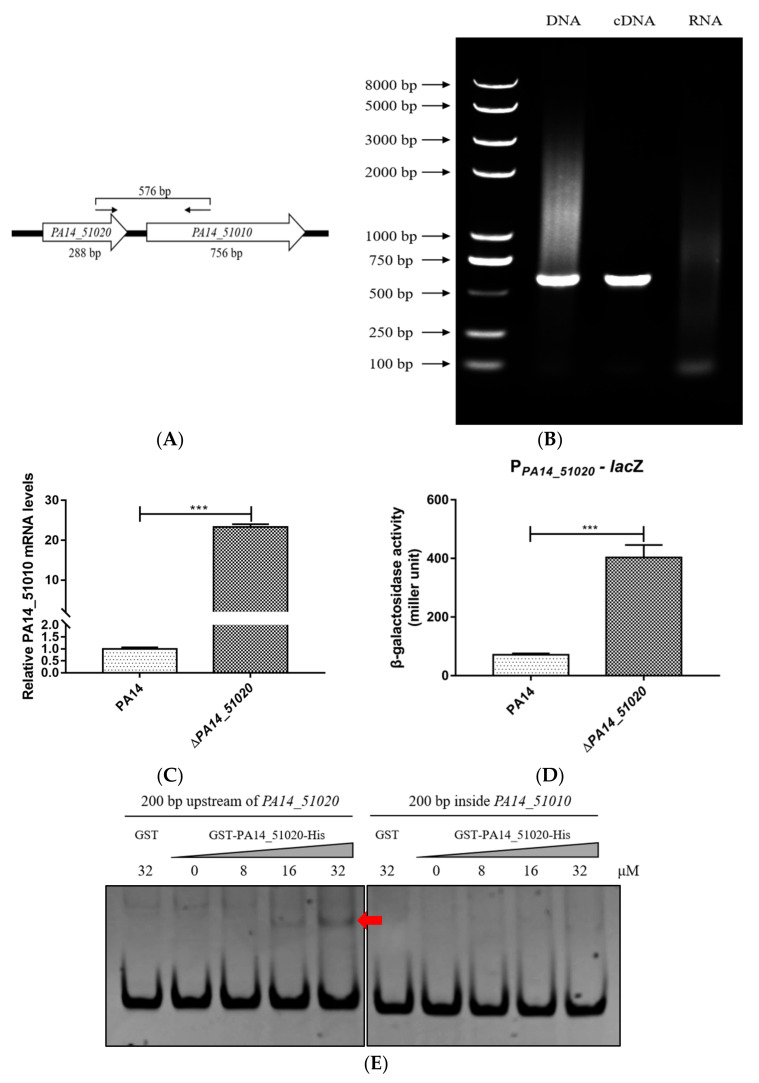
PA14_51020 regulates the operon of *PA14_51020* and *PA14_51010*. (**A**) Sketch map of the *PA14_51020-PA14_51010* operon. Arrows indicate the location and direction of primers used in RT-PCR. Sequences of the primers (*PA14_51020-F* and *PA14_51010-R*) were listed in Table 1. The size of the PCR product is 576 bp. (**B**) Total RNA was isolated from PA14, followed by the synthesis of cDNA. The cDNA was used as the template in PCR. The chromosomal DNA and total RNA were used as a positive control and a negative control, respectively. (**C**) Wild-type PA14 and the Δ*PA14_51020* mutant were grown to an OD_600_ of 1.0, followed by RNA isolation. The mRNA levels of *PA14_51010* were determined by quantitative RT-PCR. The ribosomal protein coding gene *rpsL* was used as an internal control. (**D**) The β-galactosidase activity in PA14 and the Δ*PA14_51020* mutant carrying the P*_PA14_51020_*-*lacZ* transcriptional fusion. Data represent the results from three independent experiments. ***, *p* < 0.001, by Student’s *t* test. (**E**) Purified GST-PA14_51020-His protein was incubated with 20 ng of the DNA fragment of the promoter region of the *PA14_51020-PA14_51010* operon or inside the *PA14_51010* open reading frame. The DNA bands were observed by ethidium bromide staining. The purified GST-His protein was used as a negative control. The arrow indicates the protein–DNA probe complex.

**Figure 3 microorganisms-09-00753-f003:**
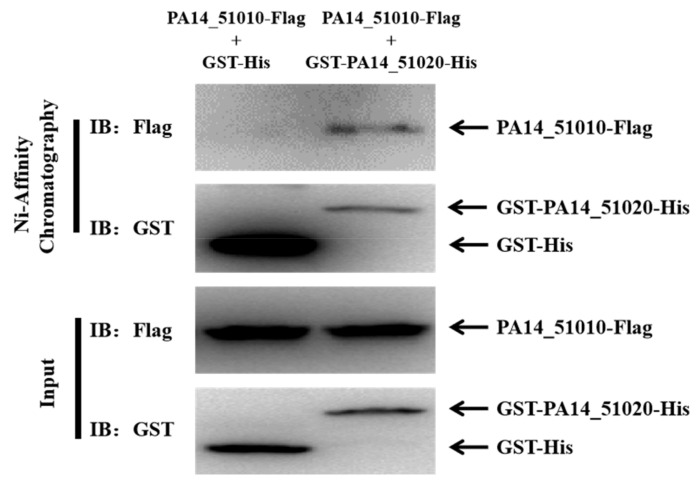
Interaction between PA14_51020 and PA14_51010. DH5α cells carrying pMMB67EH-GST-His or pMMB67EH-GST-*PA14_51020*-His were grown to an OD_600_ of 0.6 and then grown in the presence of 1 mM isopropyl-β-d-thiogalactoside (IPTG) for 4 h. The bacteria were lysed and subjected to chromatography with Ni-NTA beads. The bound protein was eluted by the elution buffer. The lysate of DH5α cells expressing the pMMB67EH-*PA14_51010*-Flag was mixed with same amount of the purified GST-His or GST-PA14_51020-His. Then, the mixtures were subjected to chromatography with Ni-NTA beads. The Flag-tagged PA14_51010, GST-tagged PA14_51020 and GST were detected by Western blot.

**Figure 4 microorganisms-09-00753-f004:**
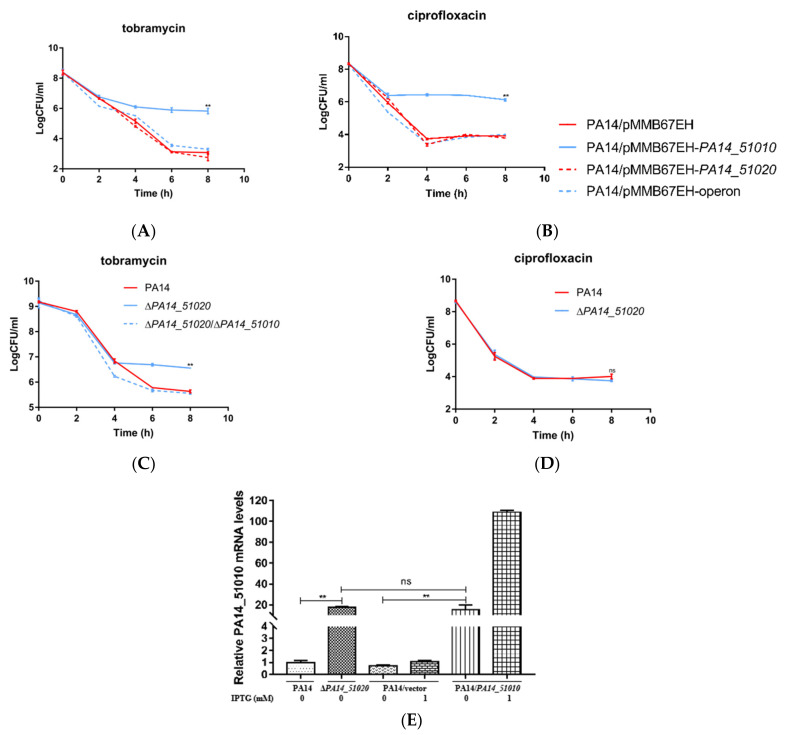
PA14_51020 and PA14_51010 regulate persister formation. (**A**,**B**) Wild-type PA14 carrying pMMB67EH, pMMB67EH-*PA14_51010*, pMMB67EH-*PA14_51020* or the pMMB67EH-operon (*PA14_51020-PA14_51010*) were treated with 5 µg/mL tobramycin (**A**) or 0.5 µg/mL ciprofloxacin (**B**). At indicated time points the live bacteria numbers were determined by serial dilution and plating assay. (**C**) Wild-type PA14 and the Δ*PA14_51020* and Δ*PA14_51020*Δ*PA14_51010* mutants were treated with 5 µg/mL tobramycin. At indicated time points the live bacteria numbers were determined by serial dilution and plating. (**D**) Wild-type PA14 and the Δ*PA14_51020* mutant were treated with 0.5 µg/mL ciprofloxacin. At indicated time points the live bacteria numbers were determined by serial dilution and plating. **, *p* < 0.01 compared to the other samples by Student’s *t* test. (**E**) Total RNAs were isolated from the indicated strains at an OD_600_ of 1.0. The mRNA levels of PA14_51010 were determined by quantitative RT-PCR. The ribosomal protein coding gene *rpsL* was used as an internal control. Data represent the results from three independent experiments. **, *p* < 0.01, ns, not significant, by Student’s *t* test.

**Figure 5 microorganisms-09-00753-f005:**
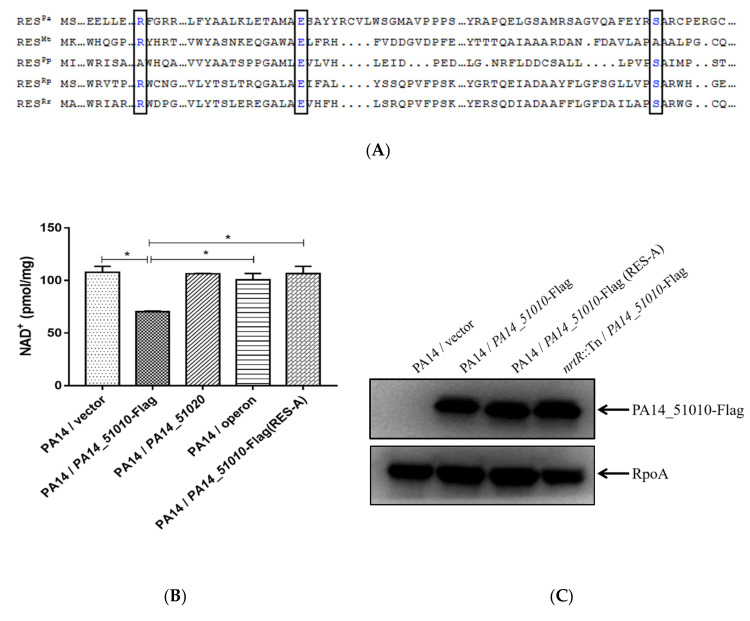
PA14_51010 promotes persister formation by reducing intracellular NAD^+^. (**A**) Protein sequence alignment of RES toxins from PA14 (RES^Pa^), *M. tuberculosis* (RES^Mt^), *Pseudomonas putida* KT2440 (RES^Pp^), *Rhodopseudomonas palustris* BisB18 (RES^Rp^) and *Rhodospirillum rubrum* ATCC 11170 (RES^Rr^) by Dnaman. The conserved residues are shown in blue and boxed. (**B**) Wild-type PA14 carrying pMMB67EH, pMMB67EH-*PA14_51010*, pMMB67EH-*PA14_51020*, pMMB67EH-operon or pMMB67EH-*PA14_51010* (RES-A) were cultured in LB to an OD_600_ of 1.0. The bacteria were collected by centrifugation, washed three times with phosphate buffered saline (PBS), and then lysed in an NAD^+^ extracting buffer by sonication. The relative quantity of NAD^+^ was normalized by the corresponding total protein amount. **, *p* < 0.01 by Student’s *t* test. (**C**) The levels of the PA14_51010-Flag in PA14/vector, PA14/*PA14_51010*-Flag, PA14/*PA14_51010*-Flag (RES-A) and *nrtR*::Tn/*PA14_51010*-Flag. (**D**,**E**) Wild-type PA14 carrying pMMB67EH, pMMB67EH-*PA14_51010* or pMMB67EH-*PA14_51010* (RES-A) were treated with 5 µg/mL tobramycin (**D**) or 0.5 µg/mL ciprofloxacin (**E**). At indicated time points the live bacteria numbers were determined by serial dilution and plating. Data represent the results from three independent experiments. *, *p* < 0.05; **, *p* < 0.01 compared to the other samples by Student’s *t* test.

**Figure 6 microorganisms-09-00753-f006:**
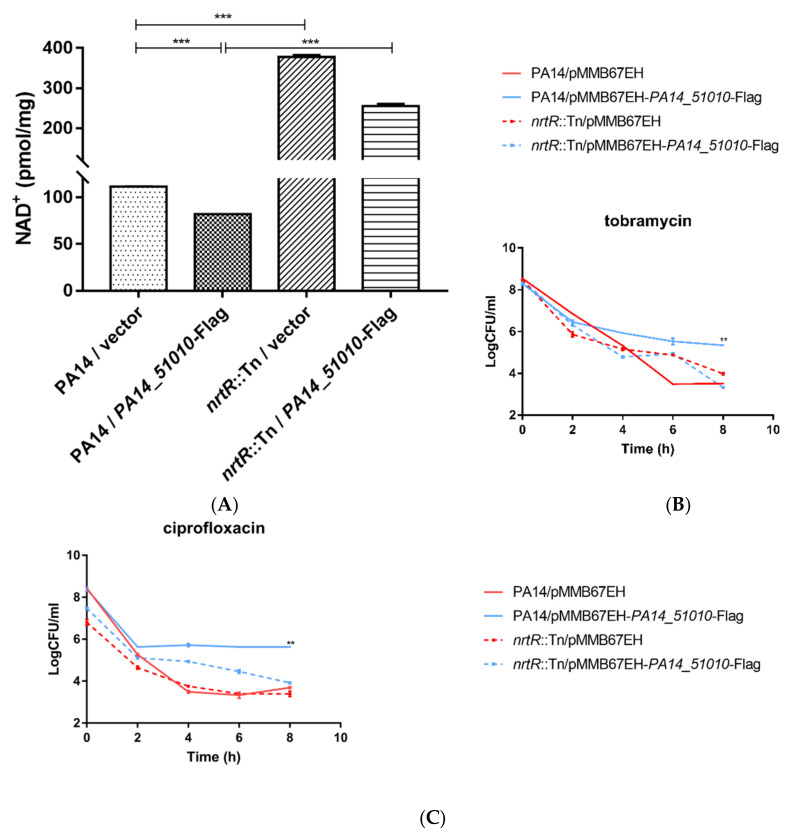
Over production of NAD^+^ counteracts the effect of overexpression of PA14_51010 in persister formation. (**A**) Wild-type PA14 and the *nrt*R::Tn mutant carrying pMMB67EH or pMMB67EH-*PA14_51010*-Flag were cultured in LB in the presence of 1 mM IPTG to an OD_600_ of 1.0. The bacteria were collected by centrifugation and washed three times with PBS. The relative quantity of NAD^+^ was normalized by the total protein amount. ***, *p* < 0.001, by Student’s *t* test. (**B**,**C**) Wild-type PA14 and the *nrtR*::Tn mutant carrying pMMB67EH or pMMB67EH-*PA14_51010* were cultured in LB in the presence of 1 mM IPTG to an OD_600_ of 1.0. The bacteria were treated with 5 µg/mL tobramycin (**B**) or 0.5 µg/mL ciprofloxacin (**C**). At indicated time points the live bacteria numbers were determined by serial dilution and plating. Data represent the results from three independent experiments. **, *p* < 0.01 compared to the other samples by Student’s *t* test.

**Table 1 microorganisms-09-00753-t001:** Bacterial strains, plasmids and primers used in this study.

Strain	Description	Source (Reference)
*P. aeruginosa*		
PA14	Wild-type strain of *Pseudomonas aeruginosa*	[32]
Δ*PA14_51020*	PA14 deleted of *PA14_51020*	This study
Δ*PA14_51010*	PA14 deleted of *PA14_51010*	This study
Δ*PA14_51020*Δ*PA14_51010*	PA14 deleted of *PA14_51020* and *PA14_51010*	This study
Plasmid		
pEX18Tc	Gene replacement vector; Tc ^r^	[33]
pMMB67EH	Expression vector with *tac* promoter; Amp ^r^	[34]
pUCP20-*lac*Z	Promoterless *lac*Z fusion vector; Amp ^r^	[35]
pMMB67EH-*PA14_51010*-Flag	*PA14_51010* gene with Flag-tag driven by *tac* promoter on pMMB67EH; Amp ^r^	This study
pMMB67EH-*PA14_40220*-Flag	*PA14_40220* gene with Flag-tag driven by *tac* promoter on pMMB67EH; Amp ^r^	This study
pMMB67EH-*PA14_21710*-Flag	*PA14_21710* gene with Flag-tag driven by *tac* promoter on pMMB67EH; Amp ^r^	This study
pMMB67EH-*PA14_28120*-Flag	*PA14_28120* gene with Flag-tag driven by *tac* promoter on pMMB67EH; Amp ^r^	This study
pMMB67EH-*PA14_28790*-Flag	*PA14_28790* gene with Flag-tag driven by *tac* promoter on pMMB67EH; Amp ^r^	This study
pMMB67EH-*PA14_60050*-Flag	*PA14_60050* gene with Flag-tag driven by *tac* promoter on pMMB67EH; Amp ^r^	This study
pMMB67EH-*PA14_71340*-Flag	*PA14_71340* gene with Flag-tag driven by *tac* promoter on pMMB67EH; Amp ^r^	This study
pMMB67EH-GST-His	GST with His-tag driven by *tac* promoter on pMMB67EH; Amp ^r^	[36]
pMMB67EH-GST-*PA14_51020*-His	*PA14_51020* gene with His-tag fused to GST driven by *tac* promoter on pMMB67EH; Amp ^r^	This study
pEX18Tc-Δ*PA14_51020*	*PA14_51020* gene of PA14 deletion on pEX18Tc; Tc ^r^	This study
pEX18Tc-Δ*PA14_51010*	*PA14_51010* gene of PA14 deletion on pEX18Tc; Tc ^r^	This study
pEX18Tc-Δ*PA14_51020*Δ*PA14_51010*	*PA14_51020* and *PA14_51010* gene of PA14 deletion on pEX18Tc; Tc ^r^	This study
pUCP20-P*_PA14_51020_*-*lac*Z	*PA14_51020* promoter of PA14 on a promoterless *lac*Z fusion vector; Amp ^r^	This study
pMMB67EH-*PA14_51020*	*PA14_51020* gene driven by *tac* promoter on pMMB67EH; Amp ^r^	This study
pMMB67EH-operon (*PA14_51020-PA14_51010*)	The operon (*PA14_51020-PA14_51010*) driven by *tac* promoter on pMMB67EH; Amp ^r^	This study
pMMB67EH-*PA14_51010* (RES-A)	*PA14_51010* gene with simultaneous replacement of the R, E, S residues with A residues driven by *tac* promoter on pMMB67EH; Amp ^r^	This study
Primer	Sequence 5′–3′	Purpose
*Eco*RI-*PA14_51020*-up-F	CCGGAATTCGCTGGAGTTGCTGACCG	*PA14_51020*, *PA14_51010* deletion
*Kpn*I-*PA14_51020*-up-R	CGGGGTACCGCCCCAATTGCTCGC	*PA14_51020*, *PA14_51010* deletion
*Kpn*I-*PA14_51020*-down-F	CGGGGTACCACGCCTGTCACGGAAAAG	*PA14_51020* deletion
*Hin*dIII-*PA14_51020*-down-F	CCCAAGCTTTCGCCGAAGCCTCTTGC	*PA14_51020* deletion
*EcoR*I-*PA14_51010*-up-F	CCGGAATTCTCAGCAGCATCCGTCGCGAT	*PA14_51010* deletion
*Kpn*I-*PA14_51010*-up-R	CGGGGTACCCTTCCGCCCCTCGCTTCCTG	*PA14_51010* deletion
*Kpn*I-*PA14_51010*-down-F	CGGGGTACCGAGCTGTTCCTGGTGG	*PA14_51020* and *PA14_51010* deletion
*Hin*dIII-*PA14_51010*-down-R	CCCAAGCTTGACGCACTTCCTCTTCC	*PA14_51020* and *PA14_51010* deletion
*Sma*I-P*_PA14_51020_*-F	TCCCCCGGGGGCAATGGGCCGATCGAATC	*PA14_51020* promoter cloning
*Bam*HI-P*_PA14_51020_*-R	CGCGGATCCCCCCAATTGCTCGCGCGCGG	*PA14_51020* promoter cloning
*Eco*RI-*PA14_51010*-F	CCGGAATTCTCCGACACCACAGGAAGCGA	*PA14_51010* cloning
*Hin*dIII-*PA14_51010*-R	CCCAAGCTTTCATTTATCATCATCATCTTTGTAATCCGCCGGATGCGGCA	*PA14_51010* cloning
*Bam*HI-*PA14_51020*-F	CGCGGATCCACGCAGCTCGAACTGGCCGG	*PA14_51020* cloning
*Hin*dIII-*PA14_51020*-R	CCCAAGCTTTCAGTGGTGGTGGTGGTGGTGGACCTTGCCGCGGATCGCAT	*PA14_51020* cloning
*Eco*RI-GST-F	CCGGAATTCTTTAAGAAGGAGATATAATGTCCCCTATACTAGGTTA	*PA14_51020* cloning
*Bam*HI-GST-R	CGCGGATCCACCAGAACCACTAGTTGAAC	*PA14_51020* cloning
*Bam*HI-*PA14_40220*-F	CGCGGATCCGCTCGTTTCACCGGTAGCGG	*PA14_40220* cloning
*Hin*dIII-*PA14_40220*-R	CCCAAGCTTCTATTTATCATCATCATCTTTGTAATCGGCGTCGCGCCGA	*PA14_40220* cloning
*Bam*HI-*PA14_21710*-F	CGCGGATCCACGCTCTGATGGGAGCGGAG	*PA14_21710* cloning
*Hin*dIII-*PA14_21710*-R	CCCAAGCTTTCATTTATCATCATCATCTTTGTAATCGCCGGTGAAGCTGGCT	*PA14_21710* cloning
*Eco*RI-*PA14_28120*-F	CCGGAATTCCCGCCAGCCTGTACGCACAA	*PA14_28120* cloning
*Bam*HI-*PA14_28120*-R	CGCGGATCCTCATTTATCATCATCATCTTTGTAATCGCCTCGCGCCAGT	*PA14_28120* cloning
*Eco*RI-*PA14_28790*-F	CCGGAATTCCAGCATATGCGGGAGCTGTT	*PA14_28790* cloning
*Hin*dIII-*PA14_28790*-R	CCCAAGCTTTCATTTATCATCATCATCTTTGTAATCGTGAGTACCAGCCC	*PA14_28790* cloning
*Eco*RI-*PA14_60050*-F	CCGGAATTCGAGCTCGGCAACCAGGCGAG	*PA14_60050* cloning
*Hin*dIII-*PA14_60050*-R	CCCAAGCTTTCATTTATCATCATCATCTTTGTAATCTCGTTGGGGCAGGT	*PA14_60050* cloning
*Eco*RI-*PA14_71340*-F	CCGGAATTCCCCCGCTCCACCCTTTCCCA	*PA14_71340* cloning
*Hin*dIII-*PA14_71340*-R	CCCAAGCTTTCATTTATCATCATCATCTTTGTAATCTTGAGGTTGCT	*PA14_71340* cloning
EMSA-upstream-F	GTTTTTCTCTCTATCACGCC	EMSA
EMSA-upstream-R	CCCCAATTGCTCGCGCGCGG	EMSA
EMSA-inside-F	CGGGCTGGAAGGTGGAGCGG	EMSA
EMSA-inside-R	CGCGGGCGTGAACAGGGCGA	EMSA
*PA14_51010*-F	GAGCCAAGCCTGTTCTAC	qRT-PCR
*PA14_51010*-R	CAGGACACAACGGTAATACG	qRT-PCR
*PA14_51020*-F	CACTCCCAACCATCAC	qRT-PCR
*PA14_51020*-R	AGGTATTCCAGCACAT	qRT-PCR

## Data Availability

Data is contained within the article.

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
