# Peer review of "Identification of a Toxin–Antitoxin System That Contributes to Persister Formation by Reducing NAD in Pseudomonas aeruginosa"

_microorganisms, 2021, doi:10.3390/microorganisms9040753_

Round 1
Reviewer 1 Report
The manuscript "Identification of a toxin-antitoxin system that contributes to persister formation by reducing NAD in Pseudomonas aeruginosa” describes the characterization of a new TA system, and demonstrate that the toxin over-expression results in reduced amount of NAD in the cells and in increased frequency of cells surviving treatment with antibiotics, the so-called persister cells.
General point:
The paper is succinctly written, and the message is clear, however the conclusion drawn by the authors, and claimed in the title, is at this stage, an overstatement. Indeed, what is shown her is only that an artificial situation, in which a toxin is over-expressed (either by expression from a plasmid or by deletion of the auto-repressor of the operon encoding the toxin) can lead to an increase in persister formation, correlated with a decrease in NAD.
If the claim in the title was true, a deletion of the TA operon should reduce the frequency of persisters, and it is not the case (Fig 4C, and lane 227).
The most interesting observation here is that modulating the NAD concentration in the cells affects the formation of persisters, and the authors should build on this.
Minor points.
- There is no doubt that the system identified here belongs to the TA family, however it is surprising that a deletion of the antitoxin from the chromosome has no effect on growth, suggesting that this toxin is no longer active in vivo. The authors must comment this point.
- Figure 3: the legend is not sufficient to understand what is shown (elution from columns, I guess, but it must be explained).
- Figure 4: A and B legend is missing. C: in the legend, it is indicated complemented strain, whereas on panel C and in the text it seems that what is shown is WT/single mutant/ double mutant.
- Ref 11 has been retracted and should not be cited anymore.
- Lanes 300-306: a recent ref showed a correlation between the level of production of a RES family toxin in M smegmatis and the reduction in [NAD], which must be much reduced to result in an effect on growth/survival (doi: 10.3390/toxins12050329)
- Lane42: a recent ref suggest to create a type VII family of TA systems (DOI: 1016/j.tim.2020.12.001 )
Author Response
The manuscript "Identification of a toxin-antitoxin system that contributes to persister formation by reducing NAD in Pseudomonas aeruginosa” describes the characterization of a new TA system, and demonstrate that the toxin over-expression results in reduced amount of NAD in the cells and in increased frequency of cells surviving treatment with antibiotics, the so-called persister cells.
General point:
The paper is succinctly written, and the message is clear, however the conclusion drawn by the authors, and claimed in the title, is at this stage, an overstatement. Indeed, what is shown her is only that an artificial situation, in which a toxin is over-expressed (either by expression from a plasmid or by deletion of the auto-repressor of the operon encoding the toxin) can lead to an increase in persister formation, correlated with a decrease in NAD.
If the claim in the title was true, a deletion of the TA operon should reduce the frequency of persisters, and it is not the case (Fig 4C, and lane 227).
Response:It has been shown that there are usually multiple TA systems on a bacterial genome. Deletion of one or multiple TA systems does not affect the persister formation, which might be due to redundancy of the TA systems. The discussion and references have been added to the manuscript.
The most interesting observation here is that modulating the NAD concentration in the cells affects the formation of persisters, and the authors should build on this.
Minor points.
- There is no doubt that the system identified here belongs to the TA family, however it is surprising that a deletion of the antitoxin from the chromosome has no effect on growth, suggesting that this toxin is no longer active in vivo. The authors must comment this point.
Response:Since we demonstrated that ResA reduces the NAD concentration, we suspect that the growth inhibitory function of ResA might be obvious under certain environmental stresses, such as carbon or amino acid starvation. The discussion has been added to the Discussion section.
- Figure 3: the legend is not sufficient to understand what is shown (elution from columns, I guess, but it must be explained).
Response:The proteins were eluted. We added the description to the figure legend.
- Figure 4: A and B legend is missing. C: in the legend, it is indicated complemented strain, whereas on panel C and in the text it seems that what is shown is WT/single mutant/ double mutant.
Response:Sorry for the mistake. The legend has been revised.
- Ref 11 has been retracted and should not be cited anymore.
Response:The reference has been replaced with a recent literature.
- Lanes 300-306: a recent ref showed a correlation between the level of production of a RES family toxin in M smegmatisand the reduction in [NAD], which must be much reduced to result in an effect on growth/survival (doi: 10.3390/toxins12050329)
- Lane42: a recent ref suggest to create a type VII family of TA systems (DOI: 1016/j.tim.2020.12.001 )
Response:The references have been added and the manuscript has been revised accordingly.
Reviewer 2 Report
The manuscript by Zhou and colleagues focuses on the effect of the PA14_51010 putative RES toxin on the persistence of Pseudomonas aeruginosa.
Comments:
A general comment is that the method section is poorly written. It is difficult to follow how the experiments were done. For instance, the authors mentioned the effect of putative toxins overexpression on bacterial growth but there is no mention of the inducer (IPTG). For RT-PCR it is not possible to conclude since we have no idea of the expected size of the PCR amplification (which primers were used?). For EMSA, there is no mention of reaction conditions (buffer, temperature, DNA concentration, competitor DNA?). For pull-down and western blot, there is no mention of protein concentrations, reaction buffer or antibodies that were used.
-Introduction
P1 line 30 Ref 1-4 are quite old and I would prefer more recent references
Line 36: paper (11) has been retracted and the authors should mention that there is massive debate about the role of TA on persistence.
Lines 40-41 TA activation by degradation of the antitoxin is a model (it has not be proven)
Lines 43-44 There are at least 79 TA in Mtb (ref 15 should be replaced by Sala et al, 2014)
Generally, there is a lack of data in the introduction of what we know about persistence in Pseudomonas.
-Results
Why do the authors chose this persistence assay?
Fig 1: Why the toxin genes are flag tagged (It would be better to use the native sequence to avoid inactivation or activation by the tag?)? What is the effect on bacterial growth of overexpressed toxin genes (in rich media, with no antibiotics added)?
P6 lines 148-149. Authors state that there is no effect of overexpression of toxin genes on survival rate whereas most of them decrease the survival rate with tobramycine more rapidly than with no overexpressed gene.
Fig 2D: what is the size and sequence of the promoter fragment that was used to construct lacZ transcriptional fusions?
Fig 2E: Is the protein PA14_51020 with a His tag functional? Why use purified GST as a control? Why not using non specific DNA?
P7 line 191: pull down are to check if T and A interact so that they could function as type II TA system
Fig4A: color legend is missing. Are the genes flag tagged as in Fig1 or are they native constructs? For the overexpression of the operon, are the genes tagged or not?
P9 line 234 The authors cannot conclude that PA14_51010-51020 form a TA because there is no evidence that PA14_51010 is a toxin and that PA14_51020 counteract its toxicity. The authors state in the discussion that changing the start codon of the putative toxin increase its toxicity but nothing is shown and there are no evidence that the putative antitoxin is able to inactivate this modified toxin.
Fig 5 There is no evidence that the RES mutant is expressed at the same level as the WT and one cannot exclude that the effects are due to mutant protein degradation or aggregation. For protein overexpression in the Pseudomonas ntr mutant, there are no data showing that the levels of proteins are the same as in the WT context.
Author Response
Comments and Suggestions for Authors
The manuscript by Zhou and colleagues focuses on the effect of the PA14_51010 putative RES toxin on the persistence of Pseudomonas aeruginosa.
Comments:
A general comment is that the method section is poorly written. It is difficult to follow how the experiments were done. For instance, the authors mentioned the effect of putative toxins overexpression on bacterial growth but there is no mention of the inducer (IPTG). For RT-PCR it is not possible to conclude since we have no idea of the expected size of the PCR amplification (which primers were used?). For EMSA, there is no mention of reaction conditions (buffer, temperature, DNA concentration, competitor DNA?). For pull-down and western blot, there is no mention of protein concentrations, reaction buffer or antibodies that were used.
Response:The concentration of IPTG and the induction time have been added to the manuscript.
The primers for the RT-PCR were listed in Table 1 and described in the figure legend. The size of the PCR product was labeled in Fig. 2A.
The experimental conditions of the EMSA and pull-down assays have been added to the manuscript.
We also added more detailed information to the Method section.
-Introduction
P1 line 30 Ref 1-4 are quite old and I would prefer more recent references
Response:We replaced the reference #2-4 with relevant literatures published in 2017 and 2019.
Line 36: paper (11) has been retracted and the authors should mention that there is massive debate about the role of TA on persistence.
Response:The reference has been replaced with a recent literature.
Lines 40-41 TA activation by degradation of the antitoxin is a model (it has not be proven)
Response:We revised the manuscript.
Lines 43-44 There are at least 79 TA in Mtb (ref 15 should be replaced by Sala et al, 2014)
Response:The reference has been replaced.
Generally, there is a lack of data in the introduction of what we know about persistence in Pseudomonas.
Response:We added more introduction on the TA systems and persistence of P. aeruginosa in the Introduction section.
-Results
Why do the authors chose this persistence assay?
Response:The bacterial killing assay is usually utilized to determine the persister level.
Fig 1: Why the toxin genes are flag tagged (It would be better to use the native sequence to avoid inactivation or activation by the tag?)? What is the effect on bacterial growth of overexpressed toxin genes (in rich media, with no antibiotics added)?
Response:Since we studied the predicted type II TA systems, we planned to examine the interaction between the toxin and antitoxins. Therefore, we constructed the FLAG tagged toxin genes for functional studies and potential pull-down assays. We agree that addition of amino acids might affect the function of a protein. The FLAG sequence is a widely used tag and we had not seen report on the interference of protein function by the tag, thus we added the tag at the C-terminus. In our assay, we did not observe obvious growth defect caused by overexpression of the predicted toxin genes.
P6 lines 148-149. Authors state that there is no effect of overexpression of toxin genes on survival rate whereas most of them decrease the survival rate with tobramycine more rapidly than with no overexpressed gene.
Response:We revised the manuscript based on the results.
Fig 2D: what is the size and sequence of the promoter fragment that was used to construct lacZ transcriptional fusions?
Response:The primers used to amplify the promoter region were listed in Table 1. The size of the PCR product is 500 bp. The description has been added in the Method section.
Fig 2E: Is the protein PA14_51020 with a His tag functional? Why use purified GST as a control? Why not using non specific DNA?
Response:To perform the EMSA, we need to purify the PA14_51020 protein, therefore we added a His tag to the protein. However, we had difficulties in purifying enough His-tagged PA14_51020 from E. coli. We then constructed a PA14_51020 fused with a gst gene at the N-terminus (gst-PA14_51020). When purified through glutathione-affinity chromatography, we observed that more than half of the full length protein was cleaved. To solve this problem, we added the His tag to the C-terminus of the gst-PA14_51020, resulting in gst-PA14_51020-His. With this construct, we were able to purify sufficient protein for the EMSA by using Ni-NTA beads. Therefore, we used GST protein as a control.
It has been reported that the antitoxin of the type II TA system binds to its own promoter, we thus used 200 bp sequence inside the PA14_51020 coding region in the EMSA as the negative control. Our results demonstrate that PA14_51020 binds to its own promoter region, indicating that the tagged protein is functional, at least in this assay.
P7 line 191: pull down are to check if T and A interact so that they could function as type II TA system
Response:We revised the manuscript as suggested by the reviewer.
Fig4A: color legend is missing. Are the genes flag tagged as in Fig1 or are they native constructs? For the overexpression of the operon, are the genes tagged or not?
Response:The color legend has been added to the figure. In Fig. 4A, all the genes are native construct without any tag. To clarify the cloned genes, those with tags were labeled with the names of the tags, e.g. FLAG, GST or His, in all the other figures.
P9 line 234 The authors cannot conclude that PA14_51010-51020 form a TA because there is no evidence that PA14_51010 is a toxin and that PA14_51020 counteract its toxicity. The authors state in the discussion that changing the start codon of the putative toxin increase its toxicity but nothing is shown and there are no evidence that the putative antitoxin is able to inactivate this modified toxin.
Response:We added the results of the modified toxin in the supplementary data.
In our study, we demonstrated that PA14_51010 and PA14_51020 genes form an operon, PA14_51020 represses the expression of the operon, the two encoded proteins bind to each other, and overexpression of PA14_51010 reduces NAD level and promotes persister formation, which is counteracted by co-expression of PA14_51020. These results suggest that the two proteins form a TA system.
Fig 5 There is no evidence that the RES mutant is expressed at the same level as the WT and one cannot exclude that the effects are due to mutant protein degradation or aggregation. For protein overexpression in the Pseudomonas ntr mutant, there are no data showing that the levels of proteins are the same as in the WT context.
Response:We examined the protein levels by western blot. The results have been added to the manuscript (Fig. 5C).
Reviewer 3 Report
The authors studies toxin antitoxin system in persister formation. The study covers an important topic. I would like the authors to highlight the novel aspects of their study as the study appears to be similar to some other studies in the area. Here are some other suggestions:
1- What is the meaning of "complemented" strain in the legend of Fig. 4?
2- Discussion is too short. The authors should expand it significantly. For example, they can cite P. aeruginosa studies in other parts of the world such as PMID: 18499192 DOI: 10.1016/j.juro.2008.03.081 and PMID: 30907313 DOI: 10.2174/1566524019666190321113008. They can also include a section on future directions and recommendations.
3- Data availability statement does not appear reasonable. Are the authors claiming that there are no other gels or data associated with this study?
4- Fig. 2E. Can the authors provide a better quality gel?
5- Fig 2B. Can the authors provide the full gel?
Author Response
Comments and Suggestions for Authors
The authors studies toxin antitoxin system in persister formation. The study covers an important topic. I would like the authors to highlight the novel aspects of their study as the study appears to be similar to some other studies in the area.
Response:We expanded the Introduction and Discussion to highlight the novelty of our study.
Here are some other suggestions:
1- What is the meaning of "complemented" strain in the legend of Fig. 4?
Response:Sorry for the mistake. It should be the ΔPA14_51010ΔPA14_51020 double mutant. The legend has been revised.
2- Discussion is too short. The authors should expand it significantly. For example, they can cite P. aeruginosa studies in other parts of the world such as PMID: 18499192 DOI: 10.1016/j.juro.2008.03.081 and PMID: 30907313 DOI: 10.2174/1566524019666190321113008. They can also include a section on future directions and recommendations.
Response:The references have been included in the manuscript and the discussion has been expanded as requested.
3- Data availability statement does not appear reasonable. Are the authors claiming that there are no other gels or data associated with this study?
Response:The statement has been revised.
4- Fig. 2E. Can the authors provide a better quality gel?
5- Fig 2B. Can the authors provide the full gel?
Response:We replaced the figures as requested.
Round 2
Reviewer 3 Report
NA
Author Response
Dear reviewer:
- Introduction: lines 52-53. P. aeruginosa infections in cystic fibrosis patients are not due to a typical immunocompromised conditions; on the other hand the mucociliary deficiency induces the establishment of an environment which promotes the P. aeruginosa growth.
Response: The manuscript has been revised as suggested.
- In Figures 1, 4 and 6 Tobramycin and Ciprofloxacin were used at 5 and 0.5 ug/ml. What are MIC values of these antibiotics? Indicate them in the text.
Response: The MIC values have been added to the manuscript (lines 178-180).
Please contact me by email if I can provide any additional information.